# Partitioning changes in ecosystem productivity by effects of species interactions in biodiversity experiments

**Jing Tao[1], Charles A Nock[2], Eric B Searle[3], Shongming Huang[4], Rongzhou Man[3]\*, Hua Yang[5]\*, Grégoire T Freschet[6], Cyrille Violle[7], Ji Zheng[8]**

[1]Jilin Provincial Academy of Forestry Sciences, Changchun, China; [2]Department of Renewable Resources, Faculty of Agriculture, Life, and Environmental Sciences, University of Alberta, Edmonton, Canada; [3]Ontario Ministry of Natural Resources and Forestry, Ontario Forest Research Institute, Sault Ste. Marie, Canada; [4]Forestry Division, Department of Agriculture and Forestry, Government of Alberta, Edmonton, Canada; [5]College of Forestry, Beijing Forestry University, Beijing, China; [6]Station d'Ecologie Théorique et Expérimentale, CNRS, Foix, France; [7]CEFE, CNRS, EPHE, IRD, Montpellier, France; [8]School of Agriculture and Biology, and Shanghai Urban Forest Ecosystem Research Station of National Forestry and Grassland Administration, Shanghai Jiao Tong University, Shanghai, China

**\*For correspondence:**
rongzhou.man@ontario.ca (RM);
huayang8747@163.com (HY)

**Competing interest:** The authors declare that no competing interests exist.

## eLife Assessment

The authors propose that positive biodiversity-ecosystem functioning relationships found in experiments have been exaggerated because commonly used statistical analyses are flawed. To remedy this, a new type of analysis based on a concept of "partial density monoculture yield" is proposed. However, the presented concept and analysis methods are not reproducibly described (how can partial density monoculture yield experimentally be assessed?), do not appear to be complete, and are **inadequate** for hypothesis testing. The reviewers found that the authors misinterpret current research in the field and made limited efforts to understand or address the reviewer comments about this study.

**Abstract** Species interactions affect ecosystem productivity. Positive interactions (resource partitioning and facilitation) increase productivity while negative interactions (species interference) decrease productivity relative to the null expectations defined by monoculture yields. Effects of competitive interactions (resource competition) can be either positive or negative. Distinguishing effects of species interactions is therefore difficult, if not impossible, with current biodiversity experiments involving mixtures and full density monocultures. To partition changes in ecosystem productivity by effects of species interactions, we modify null expectations with competitive growth responses, i.e., proportional changes in individual size (biomass or volume) expected in mixture based on species differences in growth and competitive ability. We use partial density (species density in mixture) monocultures and the competitive exclusion principle to determine maximum competitive growth responses and full density monoculture yields to measure species ability to achieve maximum competitive growth responses in mixture. Deviations of observed yields from competitive expectations represent the effects of positive/negative species interactions, while the differences between competitive and null expectations reflect the effects of competitive interactions. We demonstrate the effectiveness of our competitive partitioning model in distinguishing effects of species interactions using both simulated and experimental species mixtures. Our

competitive partitioning model enables meaningful assessments of species interactions at both species and community levels and helps disentangle underlying mechanisms of species interactions responsible for changes in ecosystem productivity and identify species mixtures that maximize positive effects.

## Introduction

Detecting changes in ecosystem productivity with biodiversity and understanding how these changes are affected by mechanisms of species interactions has been a central focus in ecology for decades (*Loreau, 2000*; *Loreau and Hector, 2019*; *Mahaut et al., 2020*). However, effective methods are lacking (*Barry et al., 2019*; *Bourrat et al., 2023*; *Drake, 2003*; *Forrester and Pretzsch, 2015*; *Jaillard et al., 2018*; *Loreau and Hector, 2019*). The method of additive partitioning was developed from the covariance equation to quantify changes in ecosystem productivity by comparing species performance in mixture with those expected from their monocultures using a replacement design (*Barry et al., 2019*; *Loreau and Hector, 2001*). The resultant changes in ecosystem productivity (commonly referred to as net biodiversity effects) are mathematically decomposed into additive components, namely the complementarity effect (CE) measuring community average response and the selection effect (SE) measuring interspecies differences (*Forrester and Pretzsch, 2015*; *Loreau and Hector, 2001*; *Loreau and Hector, 2019*). CE and SE are widely used to examine mechanisms of species interactions responsible for biodiversity effects, i.e., positive CE for effects of positive interactions and positive SE for effects of competitive interactions (*Barry et al., 2019*; *Feng et al., 2022*; *Hagan et al., 2021*; *Loreau and Hector, 2001*; *Montès et al., 2008*; *Petchey, 2003*; *Polley et al., 2003*), despite that neither of the additive components represents specific effects of species interactions (*Bourrat et al., 2023*; *Forrester and Pretzsch, 2015*; *Hagan et al., 2021*; *Hooper et al., 2005*; *Loreau et al., 2012*; *Loreau and Hector, 2019*; *Montès et al., 2008*; *Petchey, 2003*).

Among the interspecific interactions in plants, positive interactions (resource partitioning and facilitation or collectively referred to as complementarity) increase productivity and negative interactions (species interference or interference competition) decrease productivity relative to the null expectations, i.e., species yields expected in mixture based on composition (relative yield) and full density monoculture yields (*Barry et al., 2019*; *Hooper et al., 2005*; *Loreau, 2000*; *Loreau and Hector, 2001*). Uncertainty comes with effects of competitive interactions (i.e. resource competition) that are positive for more competitive species and negative for less competitive species (*Aschehoug et al., 2016*; *Barry et al., 2019*). For example, when a highly competitive and less competitive species are mixed, more resources flow to the more competitive species. The yield gain of more competitive species from increases in resource availability (resources per capita) often exceeds the yield loss of less competitive species from decreases in resource availability (*Huston, 1997*; *Montazeaud et al., 2018*; *Pillai and Gouhier, 2019*; *Tilman et al., 1997*; *Wardle, 1999*), resulting in positive deviation from the total expected yield of both species (i.e. positive biodiversity effect). Competitive interactions can also result in negative biodiversity effects if the less competitive species suffers more than the more competitive species benefits.

The positive contribution of competitive dominance to ecosystem productivity based on the null expectation can be illustrated with species mixture data. In the situations of both simulated and experimental mixtures with two species (*Huang et al., 2009*; *Mahaut et al., 2020*), the partial density monoculture yield of the more competitive species exceeded the total expected yield of both species in 32 of the 35 mixtures (*Table 1*), meaning that a higher productivity expected for a given species mixture under the null hypothesis can be generally achieved by removing less competitive species from the mixture. In other words, the positive biodiversity effects of mixtures detected through the null expectation can be obtained from the monocultures of more competitive species at partial (lower) densities. These positive biodiversity effects, however, result from competition-induced changes in resource availability, i.e., the yield gain of more competitive species from all resources in the mixture exceeds the total yield loss expected for the less competitive species and is not related to effects of positive species interactions. Competitive interactions are the predominant type of interspecific relationships in plants and occur in all mixtures where constituent species are competitively different (*Goldberg and Barton, 1992*; *Pillai and Gouhier, 2019*). Reporting positive biodiversity effects,

**Table 1.** Simulated and experimental species mixtures.

Biodiversity effects with partial density monocultures (BPM) are calculated from the differences between the partial density monoculture yield of more competitive species and total mixture yield expected from species relative yields and full density monoculture yields (see competitive exclusion in *Supplementary file 1A and B* for detailed calculations).

| Species mixture (species1:species2) | Age | Total density | Mixture composition | | Full density monoculture yields | | Observed yields in mixtures | | Partial density monoculture yields | | |
|---|---|---|---|---|---|---|---|---|---|---|---|
| | | | Species1 | Species2 | Species1 | Species2 | Species1 | Species2 | Species1 | Species2 | BPM |
| Simulated mixed trembling aspen and white spruce in western Canada generated with GYPSY model (*Huang et al., 2009*) (density and stand volume yield, stems and m³ per hectare) | | | | | | | | | | | |
| Populus:Picea | 20 years | 11,000 | 0.9 | 0.1 | 23.7 | 0.7 | 22.3 | 0.1 | 23.3 | 0.5 | 1.9 |
| Populus:Picea | 20 years | 11,000 | 0.7 | 0.3 | 23.7 | 0.7 | 19.0 | 0.2 | 22.5 | 0.6 | 5.7 |
| Populus:Picea | 20 years | 11,000 | 0.5 | 0.5 | 23.7 | 0.7 | 15.1 | 0.3 | 21.4 | 0.6 | 9.2 |
| Populus:Picea | 20 years | 11,000 | 0.3 | 0.7 | 23.7 | 0.7 | 10.6 | 0.4 | 19.8 | 0.7 | 12.2 |
| Populus:Picea | 20 years | 11,000 | 0.1 | 0.9 | 23.7 | 0.7 | 5.0 | 0.5 | 16.9 | 0.7 | 13.9 |
| Populus:Picea | 40 years | 3,600 | 0.9 | 0.1 | 110.4 | 27.2 | 104.5 | 3.4 | 108.3 | 15.5 | 6.3 |
| Populus:Picea | 40 years | 3,600 | 0.7 | 0.3 | 110.4 | 27.2 | 90.3 | 7.6 | 103.5 | 20.7 | 18.1 |
| Populus:Picea | 40 years | 3,600 | 0.5 | 0.5 | 110.4 | 27.2 | 72.4 | 12.1 | 97.0 | 23.4 | 28.2 |
| Populus:Picea | 40 years | 3,600 | 0.3 | 0.7 | 110.4 | 27.2 | 49.6 | 17.5 | 87.4 | 25.3 | 35.2 |
| Populus:Picea | 40 years | 3,600 | 0.1 | 0.9 | 110.4 | 27.2 | 20.6 | 23.9 | 70.6 | 26.7 | 35.1 |
| Populus:Picea | 60 years | 1,800 | 0.9 | 0.1 | 189.7 | 88.3 | 180.5 | 9.6 | 186.1 | 47.0 | 6.6 |
| Populus:Picea | 60 years | 1,800 | 0.7 | 0.3 | 189.7 | 88.3 | 158.1 | 22.4 | 177.3 | 64.3 | 18.0 |
| Populus:Picea | 60 years | 1,800 | 0.5 | 0.5 | 189.7 | 88.3 | 128.1 | 38.1 | 164.8 | 73.7 | 25.8 |
| Populus:Picea | 60 years | 1,800 | 0.3 | 0.7 | 189.7 | 88.3 | 85.4 | 58.5 | 144.8 | 80.5 | 26.1 |
| Populus:Picea | 60 years | 1,800 | 0.1 | 0.9 | 189.7 | 88.3 | 28.6 | 83.0 | 109.2 | 85.9 | 10.8 |
| Populus:Picea | 80 years | 1,100 | 0.9 | 0.1 | 240.2 | 143.1 | 229.6 | 13.7 | 235.6 | 73.3 | 5.1 |
| Populus:Picea | 80 years | 1,100 | 0.7 | 0.3 | 240.2 | 143.1 | 203.7 | 33.0 | 224.1 | 102.0 | 13.1 |
| Populus:Picea | 80 years | 1,100 | 0.5 | 0.5 | 240.2 | 143.1 | 167.3 | 58.2 | 207.6 | 117.8 | 15.9 |
| Populus:Picea | 80 years | 1,100 | 0.3 | 0.7 | 240.2 | 143.1 | 110.3 | 94.0 | 179.1 | 129.4 | 6.8 |
| Populus:Picea | 80 years | 1,100 | 0.1 | 0.9 | 240.2 | 143.1 | 26.2 | 139.6 | 124.5 | 138.9 | −28.4 |
| Experimental grassland mixtures (density and aboveground biomass yield, stems, and grams per pot) (*Mahaut et al., 2020*) | | | | | | | | | | | |
| Bromus:Dactylis | 13 weeks | 6 | 0.5 | 0.5 | 5.5 | 9.8 | 2.6 | 8.6 | 5.3 | 11.5 | 3.8 |
| Bromus:Lotus | 13 weeks | 6 | 0.5 | 0.5 | 5.5 | 6.3 | 3.8 | 3.2 | 5.3 | 4.7 | −1.2 |
| Bromus:Plantago | 13 weeks | 6 | 0.5 | 0.5 | 5.5 | 8.6 | 1.9 | 10.9 | 5.3 | 16.1 | 9.0 |
| Bromus:Sanguisorba | 13 weeks | 6 | 0.5 | 0.5 | 5.5 | 4.4 | 2.9 | 2.2 | 5.3 | 3.6 | 0.3 |
| Bromus:Trifolium | 13 weeks | 6 | 0.5 | 0.5 | 5.5 | 13.3 | 3.8 | 9.3 | 5.3 | 14.2 | 4.8 |
| Dactylis:Lotus | 13 weeks | 6 | 0.5 | 0.5 | 9.8 | 6.3 | 11.8 | 2.0 | 11.5 | 4.7 | 3.4 |
| Dactylis:Plantago | 13 weeks | 6 | 0.5 | 0.5 | 9.8 | 8.6 | 4.4 | 9.5 | 11.5 | 16.1 | 2.3 |
| Dactylis:Sanguisorba | 13 weeks | 6 | 0.5 | 0.5 | 9.8 | 4.4 | 6.0 | 1.9 | 11.5 | 3.6 | 4.4 |
| Dactylis:Trifolium | 13 weeks | 6 | 0.5 | 0.5 | 9.8 | 13.3 | 7.5 | 7.7 | 11.5 | 14.2 | 2.6 |
| Lotus:Plantago | 13 weeks | 6 | 0.5 | 0.5 | 6.3 | 8.6 | 1.0 | 16.2 | 4.7 | 16.1 | 8.6 |
| Lotus:Sanguisorba | 13 weeks | 6 | 0.5 | 0.5 | 6.3 | 4.4 | 1.9 | 2.6 | 4.7 | 3.6 | −0.6 |
| Lotus:Trifolium | 13 weeks | 6 | 0.5 | 0.5 | 6.3 | 13.3 | 2.4 | 9.2 | 4.7 | 14.2 | 4.4 |
| Plantago:Sanguisorba | 13 weeks | 6 | 0.5 | 0.5 | 8.6 | 4.4 | 11.6 | 1.1 | 16.1 | 3.6 | 9.6 |
| Plantago:Trifolium | 13 weeks | 6 | 0.5 | 0.5 | 8.6 | 13.3 | 10.2 | 6.9 | 16.1 | 14.2 | 3.3 |
| Sanguisorba:Trifolium | 13 weeks | 6 | 0.5 | 0.5 | 4.4 | 13.3 | 1.9 | 13.4 | 3.6 | 14.2 | 5.3 |

Species codes: Populus - *Populus tremuloides*, Picea - *Picea glauca*, Bromus - *Bromus erectus*, Dactylis - *Dactylis glomerata*, Lotus - *Lotus corniculatus*, Plantago - *Plantago lanceolata*, Sanguisorba - *Sanguisorba minor*, and Trifolium - *Trifolium repens*.

without indication of possible contributions from competitive interactions, can lead to unrealistic expectations for effects of positive interactions.

Given the limits of the additive partitioning in deciphering mechanisms of changes in ecosystem productivity (*Forrester and Pretzsch, 2015*; *Loreau and Hector, 2001*; *Petchey, 2003*), some alternative methods have been explored to identify underlying mechanisms. This includes the use of metrics such as relative yield totals (*Hector, 1998*; *Roscher and Schumacher, 2016*) and transgressive overyielding or superyielding (*Drake, 2003*; *Forrester and Pretzsch, 2015*; *Špaèková and Lepš, 2001*), the multiplicative partitioning or statistical analyses to separate effects of species interactions and species composition (*Chen et al., 2020*; *Jaillard et al., 2018*; *Kirwan et al., 2009*), the identification of relative species influences via measurement of competitive interactions (*Brun et al., 2022*; *Mahaut et al., 2020*; *Montès et al., 2008*), or influences of species diversity, composition, and density on net biodiversity effects (*Jaillard et al., 2018*; *Mahaut et al., 2020*; *Polley et al., 2003*; *Roscher and Schumacher, 2016*; *Tatsumi and Loreau, 2023*). However, although these methods may indicate the presence of strong positive or competitive interactions or connections of the additive components with niche and fitness differences (*Carroll et al., 2011*; *Godoy et al., 2020*; *Turnbull et al., 2013*), they are not capable of quantifying changes in ecosystem productivity by species interactions at species or community level (*Loreau et al., 2012*).

Here, we present the competitive partitioning model, a new framework for assessing effects of species interactions on ecosystem productivity based on null expectations and competitive growth

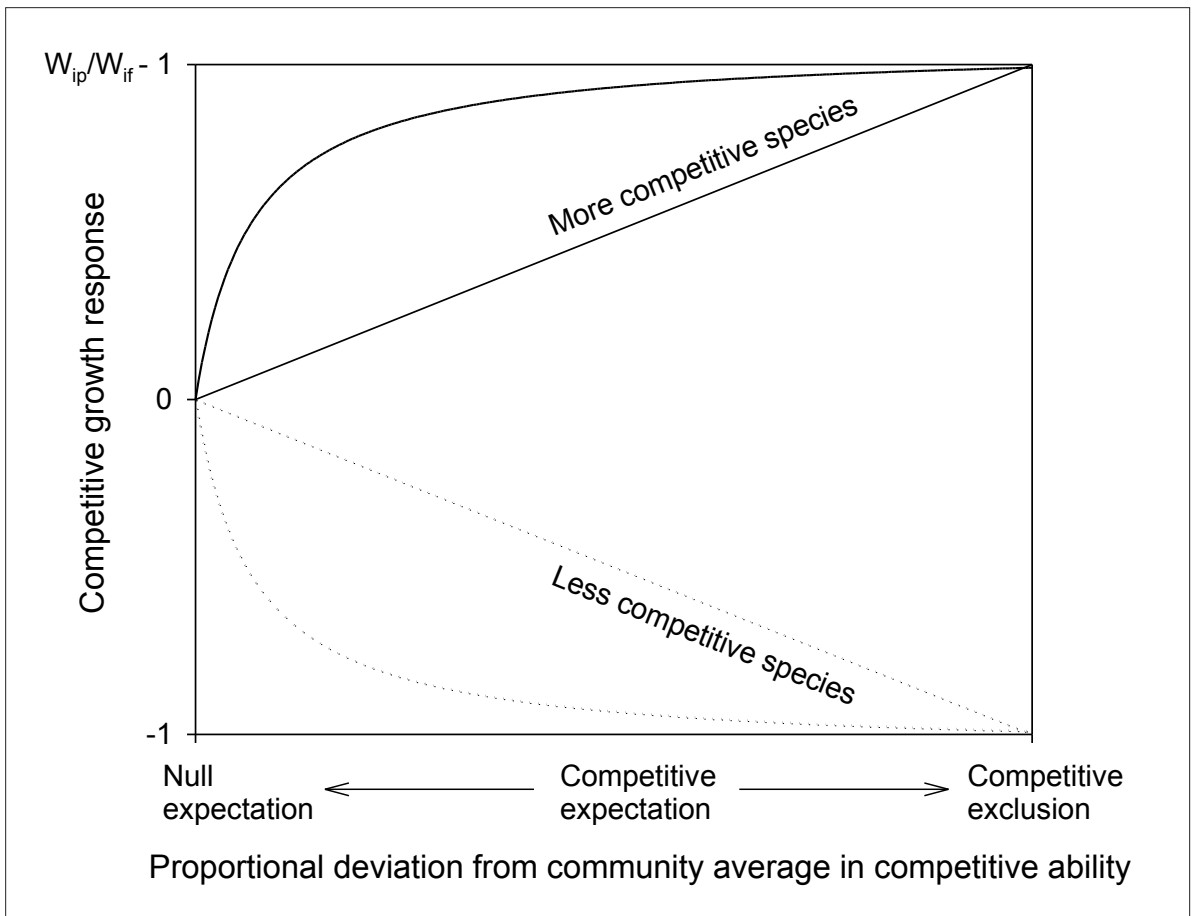

**Figure 1.** Under the competitive hypothesis, more competitive species gain size (biomass or volume) and less competitive species lose size in mixture relative to their full density monocultures. The magnitude of competitive growth responses (i.e. proportional changes in individual size) increases with relative competitive ability (proportional deviations of species competitive ability from community average) from a minimum of 0 (at community average or null expectation) to a maximum of $W_{ip}/W_{if} - 1$ for more competitive species and $-1$ for less competitive species at competitive exclusion. The changes may not be linear, greater near community average and smaller with deviations of competitive ability (*Gaudet and Keddy, 1988*) or resource availability (e.g. light, see *Brüllhardt et al., 2020*) away from community averages. $W_{if}$ represents individual size in full density monocultures, and $W_{ip}$ represents individual size in partial density monocultures.

responses, the proportional changes of individual size (biomass or volume) from full density mono-cultures to mixtures expected from species differences in growth and competitive ability (*Figure 1*). Current biodiversity experiments are generally established based on a replacement design with equal density across mixtures and monocultures (*Sackville Hamilton, 1994*) and do not provide data for estimating competitive growth responses. Thus, we applied this new methodological approach to simulated data generated with the GYPSY model (*Huang et al., 2009*) for trembling aspen (*Populus tremuloides* Michx.) and white spruce (*Picea glauca* [Moench] Voss) tree mixtures and greenhouse experimental data from grassland mixtures (*Mahaut et al., 2020*; *Table 1*). We used partial density (species density in mixture) monocultures and the competitive exclusion principle to determine maximum competitive growth responses and full density monoculture yields to measure species ability to compete for resources and achieve maximum competitive growth responses in mixture. Our objectives are to (1) demonstrate the power of this new framework in distinguishing effects of species interactions responsible for changes in ecosystem productivity, (2) illustrate how the mechanisms of changes in ecosystem productivity differ between the assessments with competitive and additive partitioning models, and (3) assess the effects of competitive interactions on ecosystem productivity relative to those of other species interactions.

## Results

### Simulated aspen and spruce mixtures

Mixed forests composed of trembling aspen (*Populus tremuloides* Michx.) and white spruce (*Picea glauca* [Moench] Voss) are widely distributed in North America (*Man and Lieffers, 1999*). The two species, occurring in pure and mixed stands of varying compositions in natural conditions, generally have different canopy structure, rooting depth, leaf phenology, and requirements for light, moisture, and nutrients and are considered different in niche (*Man and Lieffers, 1999*).

We used a growth and yield simulation system GYPSY, developed for mixed species forests in western Canada (*Huang et al., 2009*), to generate data required for determining competitive expectation, i.e., mixtures and monocultures at partial and full densities. We were interested in knowing how effects of species interactions may vary with stand age and composition. We chose four ages: 20, 40, 60, and 80 years, to represent young, middle-aged, pre-mature, and mature stands, and five relative yields from nearly pure aspen (90% aspen), aspen-dominated (70% aspen), equal mixed (50% aspen), spruce-dominated (30% aspen), to nearly pure spruce (10% aspen) mixtures at average densities on fully stocked medium productivity sites (*Peterson and Peterson, 1992*; *Table 1*). As GYPSY is developed from extensive field observations (*Huang et al., 2009*), the simulated data represent average yields in natural monocultures and mixtures of various composition and ages that may result from direct or indirect interactions between the two species (*Barry et al., 2019*; *Man and Lieffers, 1999*). Species competitive ability was approximated with stand volume in full density monocultures at site index 17 m and 14 m (height), respectively, for dominant and codominant aspen and spruce at 50 years of total age on medium productivity sites (*Alberta Forest Service, 1985*; *Huang et al., 1994*).

The partitioning of net biodiversity effects with competitive partitioning model indicated that the relative contribution of competitive interactions to net biodiversity effects decreased with age (*Figure 2a*), whereas community positive effects occurred generally in pre-mature and mature stands and more so in equal aspen and spruce mixtures (*Figure 2b*). Evidence for a facilitative effect (i.e. species observed yield in mixture >partial density monoculture yield) was only found on spruce in nearly pure mature spruce mixture (*Table 1*). Community negative effects occurred in young and middle-aged mixtures, whereas aspen did poorer than expected under the competitive expectation (*Supplementary file 1A*). Averaged across all ages and compositions, the net biodiversity effect was 14.3 $m^3$ $ha^{-1}$ or 14% of the average full density monoculture yield. On a relative basis, 57% of the value (8.2 $m^3$ $ha^{-1}$) was attributable to positive effects, 65% (9.3 $m^3$ $ha^{-1}$) to competitive effects, and –23% (–3.2 $m^3$ $ha^{-1}$) to negative effects, compared to 63% and 37% for CE and SE with additive partitioning (*Figure 2c and d*).

### Experimental grassland mixtures

*Mahaut et al., 2020*, conducted an experimental study in pots with six grassland species in full and half-density monocultures, as well as in every combination of two, three, and six species in mixture.

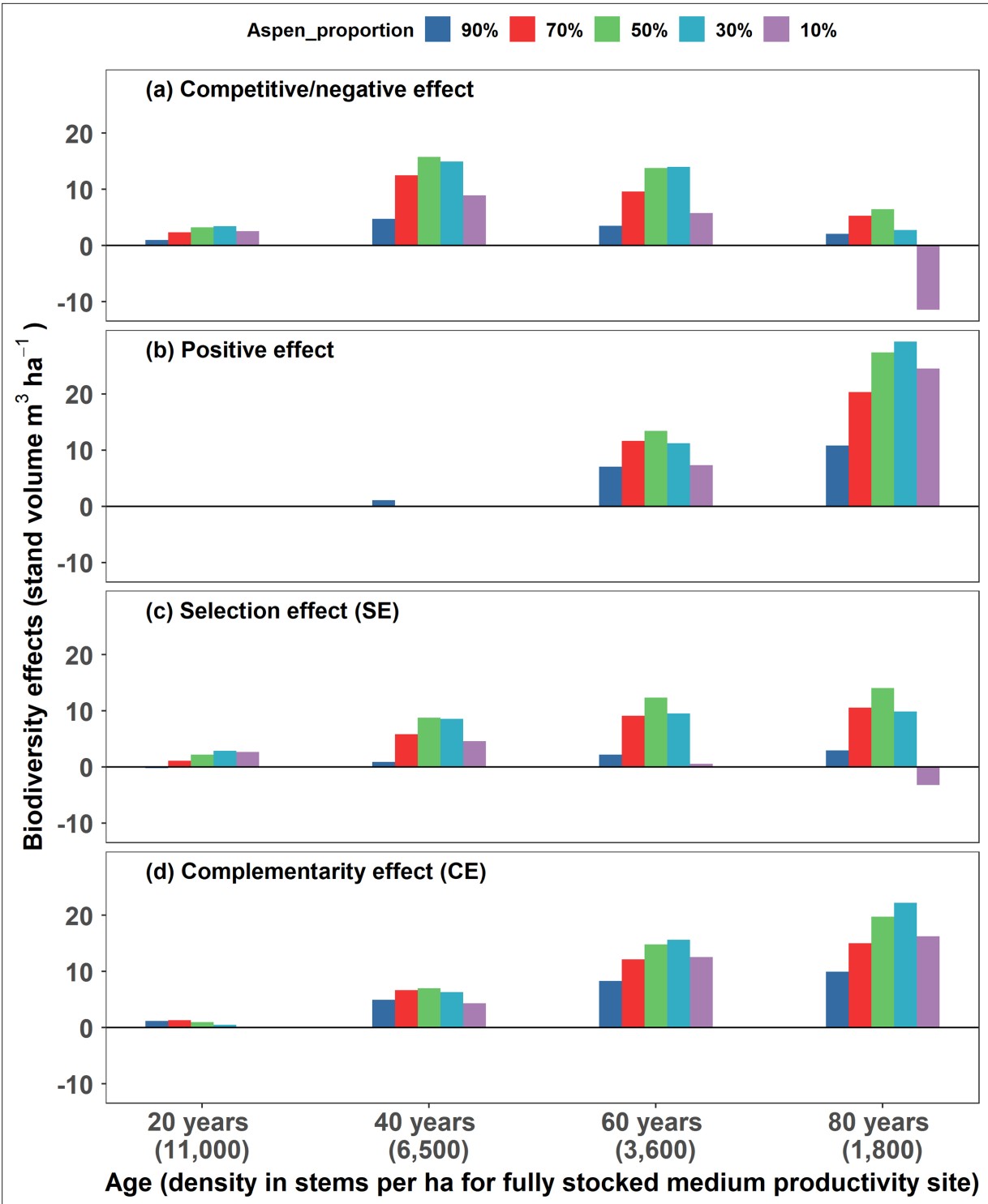

**Figure 2.** Partitioning net biodiversity effects with competitive and additive partitioning models. Partitioning net biodiversity effects (changes in stand volume in mixtures relative to full density monocultures) with competitive (**a** and **b**) and additive (**c** and **d**) partitioning models for 20- to 80-year-old mixed trembling aspen and white spruce with varying species compositions from nearly pure aspen (90% aspen) to nearly pure spruce (90% spruce) on average medium productivity sites in western Canada. The growth and yield data (*Table 1*) are generated with GYPSY (*Huang et al., 2009*), and calculations of biodiversity components with this figure are detailed in *Supplementary file 1A*.

**Table 2.** Partitioning net biodiversity effects (changes in aboveground biomass, g) with competitive and additive partitioning models using experimental data (Table 1) from a grassland diversity-productivity study (*Mahaut et al., 2020*). Calculations are detailed in *Supplementary file 1B*.

| Species mixture | Competitive partitioning | | Additive partitioning | |
|---|---|---|---|---|
| 0.50Species1: 0.50Species2 | Positive effect | Competitive/ negative effect | Complementarity effect (CE) | Selection effect (SE) |
| Bromus:Dactylis | 1.8 | 1.7 | 2.7 | 0.9 |
| Bromus:Lotus | 1.2 | –0.1 | 1.1 | –0.1 |
| Bromus:Plantago | 2.5 | 3.2 | 4.3 | 1.4 |
| Bromus:Sanguisorba | 0.1 | 0.1 | 0.1 | 0.0 |
| Bromus:Trifolium | 0.9 | 2.8 | 3.6 | 0.0 |
| Dactylis:Lotus | 4.5 | 1.2 | 4.2 | 1.6 |
| Dactylis:Plantago | 4.3 | 0.3 | 5.1 | –0.4 |
| Dactylis:Sanguisorba | 0.0 | 0.8 | 0.3 | 0.5 |
| Dactylis:Trifolium | 3.0 | 0.7 | 4.0 | –0.3 |
| Lotus:Plantago | 7.6 | 2.3 | 7.9 | 1.9 |
| Lotus:Sanguisorba | 0.0 | –0.9 | –0.6 | –0.3 |
| Lotus:Trifolium | 0.0 | 1.8 | 0.8 | 1.1 |
| Plantago:Sanguisorba | 1.5 | 4.7 | 3.9 | 2.3 |
| Plantago:Trifolium | 5.0 | 1.2 | 7.8 | –1.6 |
| Sanguisorba:Trifolium | 2.9 | 3.6 | 3.9 | 2.6 |
| Average | 2.3 | Competitive: 1.7 Negative: –0.2 | 3.3 | 0.6 |

Species codes: see **Table 1**.

The studied species vary in productivity and functional traits. We used aboveground biomass data from full and half-density monocultures, as well as all combinations of two species in full density mixtures (*Table 1*). Species competitive ability was approximated with species biomass in full density monocultures.

The competitive partitioning model suggested a weak facilitative effect in two mixtures that were associated with *Lotus corniculatus*, one of the two legume species studied (*Supplementary file 1B*). Community negative effects occurred in two mixtures with *S. minor* and between the two legumes, due to negative effects of *S. minor* on other species and *L. corniculatus* on *T. repens*. Across all 15 two-species mixtures, the average of net biodiversity effects on aboveground biomass was 3.9 g per pot or 49% of the full density monoculture yield averaged across all species. On a relative basis, 60% (2.3 g) of the value was attributable to positive effects, 45% (1.7 g) to competitive effects, and –5% (–0.2 g) to negative effects, compared to 84% and 16% for CE and SE with the additive partitioning model (*Table 2*).

## Discussion

The competitive partitioning model provides a conceptual framework that helps establish the linkage between changes in ecosystem productivity and effects of species interactions. Under this framework, in both simulated tree and experimental grassland mixtures, net biodiversity effects were generally positive, resulting from both positive and competitive interactions. The positive competitive effects result from greater competitive yield responses of more competitive species (*Huston, 1997*; *Montazeaud et al., 2018*; *Tilman et al., 1997*; *Wardle, 1999*), i.e., the yield gain of more competitive species from increases in resource availability exceeds the yield loss of less competitive species from decrease in resource availability, due to different competitive growth responses and full density monoculture yields (*Supplementary file 1A and B*). The greater competitive yield response of more competitive trembling aspen in young and middle-aged mixtures resulted in net

biodiversity effects that were predominantly from competitive interactions (*Figure 2a and b*, *Supplementary file 1A*). A facilitative effect was rarely detected, only on spruce in nearly pure spruce mature mixture, possibly due to improved nutrient availability by aspen (*Man and Lieffers, 1999*; *Peterson and Peterson, 1992*) and in mixtures with legume *L. corniculatus* in grassland mixtures, likely due to improved nitrogen availability (*Mahaut et al., 2020*). The strong negative effect of mixing on aspen (i.e. observed yields<competitive expectations, see *Supplementary file 1A*) in young and middle-aged mixtures suggests that trembling aspen originating naturally in high density does not benefit from mixing with white spruce. This is in contrast with white spruce where the deviations of observed yields from competitive expectations were generally positive, likely due to its shade tolerance and positive effects of aspen on nutrient availability (*Man and Lieffers, 1999*; *Peterson and Peterson, 1992*).

The net biodiversity effects reveal overall community yield changes from what would be expected from full density monoculture yields (*Loreau and Hector, 2001*; *Loreau and Hector, 2019*). However, the underlying mechanisms of species interactions for detected changes are uncertain, due to effects of competitive interactions that can be positive or negative. In both simulated mixed forests and experimental grassland mixtures, the partial density monoculture yield of the more competitive species exceeded the total expected yield of both species in nearly all mixtures, resulting in positive biodiversity effects when positive species interactions are not involved. Such positive biodiversity effects may help understand changes in ecosystem productivity detected through the null expectation (*Carroll et al., 2011*; *Godoy et al., 2020*) but do not support the general expectation that positive changes in ecosystem productivity are associated with positive species interactions (*Barry et al., 2019*; *Drake, 2003*; *Hooper et al., 2005*; *Loreau, 2000*) or crop mixing in agriculture and forestry where interests are positive effects (*Forrester and Pretzsch, 2015*; *Man and Lieffers, 1999*; *Montazeaud et al., 2018*). Thus, partitioning net biodiversity effects by effects of species interactions (i.e. positive, competitive, negative effects) allows for estimating relative importance of mechanisms of species interactions responsible for changes in ecosystem productivity. This enables ecologists to determine which communities are truly benefitting from diversity in the form of positive interactions and which result from interspecies differences in growth and competitive ability. In the aspen and spruce mixtures examined, community positive effects occurred largely in pre-mature and mature mixtures (≥60 years), even though net biodiversity effects and CE were positive at nearly all stages of stand development (*Figure 2*). In grassland mixtures, two mixtures (i.e. *D. glomerata - S. minor* and *L. corniculatus - T. repens*) had positive net biodiversity effects that resulted entirely from competitive interactions (*Supplementary file 1B*). These positive net biodiversity effects may indicate evidence for positive changes in ecosystem productivity when these changes were actually driven by differential outcomes of interspecific competition.

We demonstrate with simulated and experimental species mixtures that competitive interaction is a major source of the net biodiversity effects and affects the additive components of both CE and SE. In the scenario of partial density monocultures emulating a simplified scenario of competitive exclusion where positive net biodiversity effects resulted entirely from competitive interactions, CE was generally negative in tree mixtures where relative yield changes were smaller with more competitive species (negative mean relative yield change) but positive in grassland mixtures where relative yield changes were greater with more competitive species (positive mean relative yield change; *Supplementary file 1A and B*). In aspen-spruce mixtures, SE was either larger (age 80) or smaller (age 20, 40, and 60) than competitive effect. In the grassland communities, SE was similar to competitive effect in 3 of the 15 mixtures (*B. erectus - L. corniculatus*, *D. glomerata - L. corniculatus*, and *L. corniculatus - S. minor*) but substantially smaller in the remaining mixtures (*Table 2*). Even in the extreme cases where SE was nearly zero (*B. erectus - S. minor* and *B. erectus - T. repens*), competitive effect was still relatively strong, suggesting that a common assumption to attribute positive net biodiversity effects to positive species interactions when SE is close to zero (*Feng et al., 2022*; *Loreau and Hector, 2001*) is not justified. Competitive interactions amplify interspecies differences and therefore SE, while species interference on more competitive species (*B. erectus - S. minor*) and greater positive species interactions on less competitive species (*B. erectus - T. repens*) (*Supplementary file 1B*) reduce SE. Our findings, along with some recent suggestions (*Bourrat et al., 2023*; *Carroll et al., 2011*; *Godoy et al., 2020*; *Turnbull et al., 2013*), do not support some common exercises that attribute positive/negative CEs to positive/negative interactions (*Barry et al., 2019*; *Fargione et al., 2007*; *Feng et al.,*

*2022*; *Hagan et al., 2021*; *Loreau and Hector, 2001*; *Petchey, 2003*) and positive SEs to competitive interactions (*Loreau, 2000*; *Loreau and Hector, 2001*; *Montazeaud et al., 2018*).

Several assumptions are adopted with the development of the competitive partitioning model. First, we assumed that more productive species are more competitive, which was mostly true with the species mixtures examined in terms of species yields in mixture relative to their full density monoculture yields (*Table 1*). There were exceptions with *Dactylis*:*Plantago* and *Plantago*:*Trifolium* in grassland mixtures where less productive species Plantago based on full density monoculture yields performed better in mixtures, which may suggest positive effects on Plantago and negative effects on the other species. However, species yields in partial versus full density monocultures indicate that these exceptions resulted from the stagnation of more productive Plantago in full (high) density monocultures, due to restricted rooting volumes and supplies of water and nutrients in pots (*Chung et al., 2021*). The release of growth stagnation in partial density monocultures and mixtures affects not only the mechanisms of changes in ecosystem productivity assessed with competitive and additive partitioning models but also the magnitudes of the changes in ecosystem productivity detected through the null expectation (*Loreau and Hector, 2001*). In this case, the use of partial density monoculture yields in assessment of species competitive ability is probably more appropriate, which would increase the values of competitive effects and reduce those of positive effects based on the competitive expectations. Partial density monocultures help detect competition-related yield changes, which would be otherwise unknown. Second, we used partial density monocultures to determine maximum competitive growth responses in a simplified competitive exclusion where more competitive species are not affected and grow like partial density monocultures, while less competitive species are competitively eliminated. We assumed a linear relationship between competitive growth responses and species relative competitive ability (*Figure 1*), which should be a good approximation (*Gaudet and Keddy, 1988*), but may not be the most appropriate, e.g., light availability and plant height (*Brüllhardt et al., 2020*). Linear models can overestimate or underestimate competitive expectations of all species and affect individual species assessments but would have limited impacts on community level assessments, due to different directions of competitive growth responses among species.

The estimation of maximum competitive growth responses in mixtures of more than two species or in low densities may require additional efforts. In mixtures where more than one species are above average competitive ability, maximum increase in resource availability would be smaller, $\frac{1-\sum RY_{Ej}}{RY_{Ei}} - 1$ (j=species more competitive than species i), instead of $\frac{1}{RY_{Ei}} - 1$. The maximum competitive growth increases determined in partial density monocultures may need to be adjusted accordingly. For less competitive species, complete elimination would not occur in mixtures of extremely low densities or small individual size where growth is largely independent of resource availability. Proportional growth reduction ($\frac{w_{if}}{w_{ip}} - 1$, opposite of proportional growth increase in partial density monocultures $\frac{w_{ip}}{w_{if}} - 1$) or partial elimination (by more competitive species) may be more appropriate for estimating maximum competitive growth reductions.

The competitive partitioning model incorporates effects of competitive interactions into the conventional null expectation and assists: (1) understanding the mechanisms of species interactions driving positive biodiversity-productivity relationships (*Cardinale et al., 2012*; *Isbell et al., 2017*) by relative contributions of positive interactions from more species of diverse niche (*Hooper et al., 2005*; *Tilman et al., 1997*) and competitive interactions from greater yield responses of more competitive and more productive species to changes in resource availability in mixture (*Aarssen, 1997*; *Huston, 1997*; *Pillai and Gouhier, 2019*; *Tilman et al., 1997*), (2) examining the mechanisms of key species in influencing ecosystem productivity by roles of competitive interactions due to their higher productivity and competitive advantages (*Loreau and Hector, 2001*; *Mahaut et al., 2020*), (3) meaningful comparisons of changes in ecosystem productivity (i.e. relative yield totals) across different ecosystems based on competitive expectations, and (4) redefining changes in ecosystem productivity such that positive changes result from positive interactions and negative changes from negative interactions (*Drake, 2003*; *Hooper et al., 2005*; *Loreau, 2000*). Under this competitive partitioning framework, null expectations are either raised for more competitive species with increased resource availability or lowered for less competitive species with decreased resource availability in mixture relative to full density monocultures. The magnitudes of competitive growth responses are derived from species relative competitive ability in mixture and their maximum competitive growth responses determined from full and partial density monocultures under a simplified scenario of competitive

exclusion. Current biodiversity experiments do not have partial density monocultures and therefore do not provide estimates of competitive growth responses that would occur in mixture due to inter-species differences in growth and competitive ability. However, density-size/yield relationship is one of the most extensively studied areas in ecology (*Watkinson, 1980*; *Weiner and Freckleton, 2010*). Previous research can help determine maximum competitive growth responses through density-size/yield relationships (*Huang et al., 2009*). This means that the competitive partitioning model can be used to investigate changes in ecosystem productivity by effects of species interactions in current biodiversity experiments established with replacement design (*Fargione et al., 2007*; *Loreau and Hector, 2001*).

## Conclusions

Competitive interactions are the major source of biodiversity effects and affect the additive compo-nents of both CE and SE. The interpretations of CEs and SEs with specific mechanisms of species inter-actions lack mathematical or ecological justifications. For example, attributing positive biodiversity effects solely to effects of positive interactions, as commonly seen in the literature (*Barry et al., 2019*; *Fargione et al., 2007*; *Feng et al., 2022*; *Hooper et al., 2005*; *Loreau and Hector, 2001*), assumes that competitive interactions are productivity-neutral (SE close to zero) (*Feng et al., 2022*; *Loreau and Hector, 2001*). This assumption is true if ecosystem productivity is defined by total resources pool (*Tilman et al., 2014*) but generally not true if mixture productivity is defined by species monoculture yields as with the additive partitioning model (*Loreau and Hector, 2001*), due to the predominance of competitive interactions in interspecific relationships (*Goldberg and Barton, 1992*) and positive effects of competitive interactions on ecosystem productivity based on the null expectation.

The competitive partitioning model is based on ecological theories of interspecific relationships, competitive ability, and competitive exclusion principle and experimental designs of both replace-ment and additive series. This new partitioning approach enables meaningful assessments of species interactions at both species and community levels and provides detailed insights into the mechanisms of species interactions that drive changes in ecosystem productivity. We believe that our framework, which is admittedly perfectible, is one promising avenue to determine effects of species interac-tions responsible for changes in ecosystem productivity, an approach that is long sought after in biodiversity-ecosystem productivity research (*Loreau et al., 2012*; *Loreau and Hector, 2019*; *Mahaut et al., 2020*).

## Materials and methods

Under the null hypothesis, species in mixture are assumed to be competitively equivalent (i.e. equal interspecific and intraspecific interactions) and species growth (or any other function) stays the same in mixture as in monocultures (*Loreau and Hector, 2001*). Deviations of observed yields (biomass or volume) from total null expectation therefore produce net biodiversity effects or changes in ecosystem productivity (*Barry et al., 2019*; *Loreau and Hector, 2001*).

$$\sum Y_{Oi} - \sum Y_{NEi} = \sum RY_{Oi}M_{if} - \sum RY_{Ei}M_{if}$$
$$= \sum \Delta RY_i M_{if} \tag{1}$$

Thus, the net biodiversity effect is the difference between total observed yields $\sum Y_{Oi}$ and total null expectation $\sum Y_{NEi}$, calculated from species composition (or relative yield) in mixture $RY_{Ei}$ and full density monoculture yield $M_{if}$.

To separate effects of competitive interactions from those of other species interactions, we would need the hypothesis that constituent species share an identical niche but differ in growth and competitive ability (i.e. absence of positive/negative interactions). Competitive growth responses, i.e., proportional changes in individual size (biomass or volume) expected from full density monocultures to mixtures under this competitive hypothesis, can be determined from species differences in growth and competitive ability (*Figure 1*). The magnitudes of competitive growth responses depend on: (1) species resource needs in full density monocultures and (2) changes in resource availability (resources per capita) from full density monocultures to full density mixtures that are related to species compo-sition and relative ability to compete for resources in mixture.

To estimate competitive growth response, we first determine the maximum competitive growth response $MG_i$ that species can reach in a scenario of competitive exclusion (**Murray, 1986**) where a more competitive species completely dominates and grows like a partial density monoculture, while a less competitive species is competitively eliminated.

$$MG_i = \frac{w_{ip}}{w_{if}} - 1, \text{for more competitive species}$$

$$= -1, \text{for less competitive species}$$

(2)

Here, $w_{ip}$ stands for individual size (biomass or volume) in partial density (species density in mixture) monoculture and $w_{if}$ is for individual size in full density monoculture; both parameters are calculated from species monoculture yields and initial densities in partial/full density monocultures. A more competitive species gains $\frac{1}{RY_{Ei}} - 1$ in resource availability and $\frac{w_{ip}}{w_{if}} - 1$ in individual size, while a less competitive species loses 100% resource availability and size in mixture relative to full density monoculture (**Figure 1**). This simplified competitive dominance applies to mixtures where constituent species differ considerably in competitive ability, especially when competition for light is intense (**Grime, 1973**; **Lieffers et al., 1993**). In most cases, however, competitive exclusion would not occur in biodiversity experiments that are controlled in density, composition, duration, and scale. More competitive species are negatively affected by interspecific competition relative to their partial density monocultures (**Holland and DeAngelis, 2009**), while less competitive species are competitively suppressed relative to their null expectations, but not eliminated (**Aarssen, 1989**; **Connolly et al., 2001**; **Mahaut et al., 2020**; **Montazeaud et al., 2018**). Other than changes in resource availability, the estimated maximum competitive growth response would include the effects of density-dependent pests, pathogens, or microclimates.

Next, we use relative competitive ability $RC_i$ to represent species ability to compete for resources and achieve their maximum competitive growth response in mixture (**Aarssen, 1989**). $RC_i$ is defined by the deviation of species competitive ability $H_i$ from community average $\bar{H}_{i\notin}$ (excluding species i) relative to community maximum $H_m$. Competitive ability is species ability to exploit resources and can be assessed by growth attributes such as full density monoculture yields in biomass or volume (**Aarssen, 1989**; **Aschehoug et al., 2016**; **Gaudet and Keddy, 1988**; **Grace, 1990**; **Montazeaud et al., 2018**).

$$RC_i = \frac{|H_i - H_{i\notin}|}{H_m}$$

(3)

Greater competitive advantages would lead to increasingly positive competitive growth responses, and larger competitive disadvantages would lead to increasingly negative competitive growth responses in mixture relative to full density monocultures.

$$Y_{CEi} = Y_{NEi} \left[ 1 + MG_i RC_i \right]$$

(4)

Species competitive expectation $Y_{CEi}$ is the yield based on null expectation $Y_{NEi}$ and competitive growth response $MG_i RC_i$ derived from species differences in growth and competitive ability. The term $MG_i RC_i$ would be positive for more competitive species $MG_i > 0$, negative for less competitive species $MG_i < 0$, and zero when species are at community average or competitively equivalent $RC_i = 0$ (**Figure 1**). At the individual species level, the deviation of observed yield $Y_{Oi}$ from the competitive expectation $Y_{CEi}$ can be positive (representing relaxation of interspecific competition from competitive expectation and therefore dominance of positive interactions), neutral (representing similar intra- and interspecific interactions or offset of positive and negative interactions), or negative (representing dominance of negative interactions). The difference between competitive and null expectations is competitive yield change resulting from competitive interactions. Isolating the facilitative effect from those of other positive interactions (i.e. resource partitioning) is also possible for individual species by comparing species' observed yield in mixture to their partial density monoculture yield, adapting the principle of additive design for biodiversity-ecosystem productivity experiments (**Sackville Hamilton, 1994**). At the community level, the difference in total yield between competitive expectations $\sum Y_{CEi}$ and null expectations $\sum Y_{NEi}$ of all species can be attributed to the competitive effect, whereas the difference between observed yields $\sum Y_{Oi}$ and competitive expectations $\sum Y_{CEi}$ can be attributed

to the dominance of positive effects when the difference >0 or to the dominance of negative effects when the difference <0. The community positive effect can be further partitioned by mechanisms of positive interactions (resource partitioning and facilitation), and facilitative effect can be classified as mutualism (+/+), commensalism (+/0), or parasitic (+/–) based on species-specific assessments.

## Acknowledgements

We thank Gordon Kayahara and Bill Parker of Ontario Ministry of Natural Resources and Forestry, Dan Kneeshaw of Université du Québec à Montréal, and four anonymous reviewers for helpful criticism and comments on previous versions of this article, and Phil Comeau of University of Alberta for advice on growth and yield models.

## Additional information

### Funding

No external funding was received for this work.

### Author contributions

Jing Tao, Data curation, Formal analysis, Writing – original draft, Writing – review and editing; Charles A Nock, Eric B Searle, Grégoire T Freschet, Cyrille Violle, Ji Zheng, Writing – original draft, Writing – review and editing; Shongming Huang, Data curation, Writing – original draft, Writing – review and editing; Rongzhou Man, Conceptualization, Supervision, Writing – original draft, Writing – review and editing; Hua Yang, Conceptualization, Writing – original draft, Writing – review and editing

### Author ORCIDs

Rongzhou Man ⓘD https://orcid.org/0000-0003-4560-5620
Ji Zheng ⓘD https://orcid.org/0000-0003-4936-1168

Joint Public Review: https://doi.org/10.7554/eLife.98073.4.sa1
Author response https://doi.org/10.7554/eLife.98073.4.sa2

## Additional files

### Supplementary files

MDAR checklist

Supplementary file 1. Simulated mixed trembling aspen and white spruce and experimental grassland mixtures. (**A**) Simulated growth and yield of mixed trembling aspen (*Populus tremuloides Michx.*) and white spruce (*Picea glauca [Moench] Voss*) by different ages and stand compositions in western Canada. (**B**) Greenhouse experiment with grassland mixtures conducted by *Mahaut et al., 2020*.

### Data availability

All data generated or analysed during this study are included in the manuscript and supporting files (*Supplementary file 1*).

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
