## [Editor Report · eLife Assessment]

The authors propose that positive biodiversity-ecosystem functioning relationships found in experiments have been exaggerated because commonly used statistical analyses are flawed. To remedy this, a new type of analysis based on a concept of "partial density monoculture yield" is proposed. However, the presented concept and analysis methods are not reproducibly described (how can partial density monoculture yield experimentally be assessed?), do not appear to be complete, and are **inadequate** for hypothesis testing. The reviewers found that the authors misinterpret current research in the field and made limited efforts to understand or address the reviewer comments about this study.

---

## [Referee Report · Joint Public Review]

This manuscript by Tao et al. reports on an effort to better specify the underlying interactions driving the effects of biodiversity on productivity in biodiversity experiments. The authors are especially concerned with the potential for competitive interactions to drive positive biodiversity-ecosystem functioning relationships by driving down the biomass of subdominant species. The authors suggest a new partitioning schema that utilizes a suite of partial density treatments to capture so-called competitive ability.

Readers are encouraged to consider the original reviews in full, which outline the strengths and weaknesses of the work:

First version: https://elifesciences.org/reviewed-preprints/98073v1/reviews

Second version: https://elifesciences.org/reviewed-preprints/98073v2/reviews

There are no further reviews for this version because the authors declined to make further improvements to their manuscript.

---

## [Author Response]

The following is the authors’ response to the previous reviews.

**Reviewer #1 (Public review):**
As a starting point, the authors discuss the so-called "additive partitioning" (AP) method proposed by Loreau & Hector in 2001. The AP is the result of a mathematical rearrangement of the definition of overyielding, written in terms of relative yields (RY) of species in mixtures relative to monocultures. One term, the so-called complementarity effect (CE), is proportional to the average RY deviations from the null expectations that plants of both species "do the same" in monocultures and mixtures. The other term, the selection effect (SE), captures how these RY deviations are related to monoculture productivity. Overall, CE measures whether relative biomass gains differ from zero when averaged across all community members, and SE, whether the "relative advantage" species have in the mixture, is related to their productivity. In extreme cases, when all species benefit, CE becomes positive.

This is not true; positive CE does not require positive RY deviations of all species. CE is positive as long as average RY deviation is greater than 0. In a 2-species mixture, for example, if the RY deviation of one species is -0.2 and that of the other species is +0.3, CE would be still positive. Positive CE can be associated with negative NE (net biodiversity effects) when more productivity species have smaller negative RY deviation compared to positive RY deviation of less productive species. Therefore, the suggestion by the reviewer “This is intuitively compatible with the idea that niche complementarity mitigates competition (CE>0)” is not correct.

When large species have large relative productivity increases, SE becomes positive. This is intuitively compatible with the idea that niche complementarity mitigates competition (CE>0), or that competitively superior species dominate mixtures and thereby driver overyielding (SE>0).

The use of word “mitigate” indicates that the effects of niche complementarity and competition are in opposite directions, which is not true with biodiversity experiments based on replacement design. We have explained this in detail in our first responses to reviewers.

However, it is very important to understand that CE and SE capture the "statistical structure" of RY that underlies overyielding. Specifically, CE and SE are not the ultimate biological mechanisms that drive overyielding, and never were meant to be. CE also does not describe niche complementarity. Interpreting CE and SE as directly quantifying niche complementarity or resource competition, is simply wrong, although it sometimes is done. The criticism of the AP method thus in large part seems unwarranted. The alternative methods the authors discuss (lines 108-123) are based on very similar principles.

Agree. However, If CE and SE are not meant to be biological mechanisms, as suggested by the reviewer, the argument “This is intuitively compatible with the idea that niche complementarity mitigates competition (CE>0), or that competitively superior species dominate mixtures and thereby driver overyielding (SE>0)” would be invalid.

Lines 108-123 are not on our method.

The authors now set out to develop a method that aims at linking response patterns to "more true" biological mechanisms.Assuming that "competitive dominance" is key to understanding mixture productivity, because "competitive interactions are the predominant type of interspecific relationships in plants", the authors introduce "partial density" monocultures, i.e. monocultures that have the same planting density for a species as in a mixture. The idea is that using these partial density monocultures as a reference would allow for isolating the effect of competition by the surrounding "species matrix".

The authors argue that "To separate effects of competitive interactions from those of other species interactions, we would need the hypothesis that constituent species share an identical niche but differ in growth and competitive ability (i.e., absence of positive/negative interactions)." - I think the term interaction is not correctly used here, because clearly competition is an interaction, but the point made here is that this would be a zero-sum game.

We did not say that competition is not an interaction.

The authors use the ratio of productivity of partial density and full-density monocultures, divided by planting density, as a measure of "competitive growth response" (abbreviated as MG). This is the extra growth a plant individual produces when intraspecific competition is reduced.Here, I see two issues: first, this rests on the assumption that there is only "one mode" of competition if two species use the same resources, which may not be true, because intraspecific and interspecific competition may differ. Of course, one can argue that then somehow "niches" are different, but such a niche definition would be very broad and go beyond the "resource set" perspective the authors adopt. Second, this value will heavily depend on timing and the relationship between maximum initial growth rates and competitive abilities at high stand densities.

True. Research findings indicate that biodiversity effect detected with AP is not constant.

The authors then progress to define relative competitive ability (RC), and this time simply uses monoculture biomass as a measure of competitive ability. To express this biomass in a standardized way, they express it as different from the mean of the other species and then divide by the maximum monoculture biomass of all species.I have two concerns here: first, if competitive ability is the capability of a species to preempt resources from a pool also accessed by another species, as the authors argued before, then this seems wrong because one would expect that a species can simply be more productive because it has a broader niche space that it exploits. This contradicts the very narrow perspective on competitive ability the authors have adopted. This also is difficult to reconcile with the idea that specialist species with a narrow niche would outcompete generalist species with a broad niche.

Competitive ability is not necessarily associated with species niche space. Both generalist and specialist species can be more productive at a particular study site, as long as they are more capable of obtaining resources from a local pool. Remember, biodiversity experiments are conducted at a site of particular conditions, not across a range of species niche space at landscape level.

Second, I am concerned by the mathematical form. Standardizing by the maximum makes the scaling dependent on a single value.

As explained in lines 370-376, the mathematical form is a linear approximation as the relationship between competitive growth responses and species relative competitive ability is generally unknow but would be likely nonlinear. Once the relationship is determined in future research, the scaling factor is not needed.

As a final step, the authors calculate a "competitive expectation" for a species' biomass in the mixture, by scaling deviations from the expected yield by the product MG ⨯ RC. This would mean a species does better in a mixture when (1) it benefits most from a conspecific density reduction, and (2) has a relatively high biomass.Put simply, the assumption would be that if a species is productive in monoculture (high RC), it effectively does not "see" the competitors and then grows like it would be the sole species in the community, i.e. like in the partial density monoculture.Overall, I am not very convinced by the proposed method.Comments on revised version:Only minimal changes were made to the manuscript, and they do not address the main points that were raised.
**Reviewer #2 (Public review):**
This manuscript by Tao et al. reports on an effort to better specify the underlying interactions driving the effects of biodiversity on productivity in biodiversity experiments. The authors are especially concerned with the potential for competitive interactions to drive positive biodiversity-ecosystem functioning relationships by driving down the biomass of subdominant species. The authors suggest a new partitioning schema that utilizes a suite of partial density treatments to capture so-called competitive ability. While I agree with the authors that understanding the underlying drivers of biodiversity-ecosystem functioning relationships is valuable - I am unsure of the added value of this specific approach for several reasons.

No responses.

Comments on revised version:The authors changed only one minor detail in response to the last round of reviews.
**Reviewer #3 (Public review):**
Summary:This manuscript claims to provide a new null hypothesis for testing the effects of biodiversity on ecosystem functioning. It reports that the strength of biodiversity effects changes when this different null hypothesis is used. This main result is rather inevitable. That is, one expects a different answer when using a different approach. The question then becomes whether the manuscript's null hypothesis is both new and an improvement on the null hypothesis that has been in use in recent decades.

Our approach adopts two hypotheses, null hypothesis that is also with the additive partitioning model and competitive hypothesis that is new. Null hypothesis assumes that inter- and intra-specie interactions are the same, while competitive hypothesis assumes that species differ in competitive ability and growth rate. Therefore, our approach is an extension of current approach. Our approach separates effects of competitive interactions from those of other species interactions, while the current approach does not.

Strengths:In general, I appreciate studies like this that question whether we have been doing it all wrong and I encourage consideration of new approaches.Weaknesses:Despite many sweeping critiques of previous studies and bold claims of novelty made throughout the manuscript, I was unable to find new insights. The manuscript fails to place the study in the context of the long history of literature on competition and biodiversity and ecosystem functioning.

We have explained in our first responses that competition and biodiversity effects are studied in different experimental approaches, i.e., additive and replacement designs. Results from one approach are not compatible with those from the other. For example, competition effect with additive design is negative but generally positive with replacement design that is used extensively in biodiversity experiments. We have considered species competitive ability, density-growth relationship, and different effects of competitive interactions between additive and replacement design, while the current method does not reflect any of those.

The Introduction claims the new approach will address deficiencies of previous approaches, but after reading further I see no evidence that it addresses the limitations of previous approaches noted in the Introduction. Furthermore, the manuscript does not reproducibly describe the methods used to produce the results (e.g., in Table 1) and relies on simulations, claiming experimental data are not available when many experiments have already tested these ideas and not found support for them.

We used simulation data, as partial density monocultures are generally not available in previous biodiversity experiments.

Finally, it is unclear to me whether rejecting the 'new' null hypothesis presented in the manuscript would be of interest to ecologists, agronomists, conservationists, or others.

Our null hypothesis is the same as the null hypothesis with the additive partitioning assuming that inter- and intra-species interactions are the same, while our competitive hypothesis assumes that species differ in competitive ability and growth rate. Rejecting null hypothesis means that inter- and intra-species interactions are different, whereas rejecting competitive hypothesis indicates existence of positive/negative species interactions. This would be interesting to everyone.

Comments on revised version:Please see review comments on the previous version of this manuscript. The authors have not revised their manuscript to address most of the issues previously raised by reviewers.

No responses.

**Recommendations for the authors:**

**Reviewer #1 (Recommendations for the authors):**
Do take reviews seriously. Even if you think the reviewers all are wrong and did not understand your work, then this seems to indicate that it was not clearly presented.
**Reviewer #2 (Recommendations for the authors):**
I can understand that the authors are perhaps frustrated with what they perceive as a basic misunderstanding of their goals and approach. This misunderstanding however, provides with it an opportunity to clarify. I believe that the authors have tried to clarify in rebutting our statements but would do better to clarify in the manuscript itself. If we reviewers, who are deeply invested in this field, don't understand the approach and its value, then it is likely that many readers will not as well.

The additive partitioning has been publicly questioned at least for serval times since the conception of the method in 2001. Our work provides an alternative.